# The IAP family member BRUCE regulates autophagosome–lysosome fusion

Petra Ebner [1], Isabella Poetsch [1], Luiza Deszcz[1], Thomas Hoffmann[2], Johannes Zuber [2,3] & Fumiyo Ikeda [1]

Autophagy has an important role in cellular homeostasis by degrading and recycling cytotoxic components. Ubiquitination is known to target cargoes for autophagy; however, key components of this pathway remain elusive. Here we performed an RNAi screen to uncover ubiquitin modifiers that are required for starvation-induced macroautophagy in mammalian cells. Our screen uncovered BRUCE/Apollon/Birc6, an IAP protein, as a new autophagy regulator. Depletion of BRUCE leads to defective fusion of autophagosomes and lysosomes. Mechanistically, BRUCE selectively interacts with two ATG8 members GABARAP and GABARAPL1, as well as with Syntaxin 17, which are all critical regulators of autophagosome–lysosome fusion. In addition, BRUCE colocalizes with LAMP2. Interestingly, a non-catalytic N-terminal BRUCE fragment that is sufficient to bind GABARAP/GABARAPL1 and Syntaxin 17, and to colocalize with LAMP2, rescues autolysosome formation in $Bruce^{-/-}$ cells. Thus, BRUCE promotes autolysosome formation independently of its ubiquitin-conjugating activity and is a regulator of both macroautophagy and apoptosis.

[1] Institute of Molecular Biotechnology of the Austrian Academy of Sciences (IMBA), Vienna BioCenter (VBC), 1030 Vienna, Austria. [2] Research Institute of Molecular Pathology (IMP), Vienna BioCenter (VBC), 1030 Vienna, Austria. [3] Medical University of Vienna, Vienna BioCenter (VBC), 1030 Vienna, Austria. Correspondence and requests for materials should be addressed to F.I. (email: fumiyo.ikeda@imba.oeaw.ac.at)

Autophagy is an evolutionarily conserved fundamental process that maintains cellular homeostasis[1–3]. Three autophagy pathways exist in mammalian cells: macroautophagy, microautophagy, and chaperone-mediated autophagy. These are all important for lysosome-dependent degradation of cargoes such as damaged organelles and protein aggregates[4]. The macroautophagy pathway (referred to as autophagy in this study) involves autophagosome formation, mediated by multiple autophagy (ATG) proteins[1,5], followed by docking and fusion with a lysosome to become an autolysosome[6], and phagocytosed cargoes are subsequently degraded by lysosomal enzymes[7]. Genetic studies in yeast have revealed essential genes for autophagy initiation and autophagosome formation, which are conserved in mammals and very well defined[1,3]. Recent studies have also uncovered various factors with important roles in autophagosome–lysosome fusion[8], including ATG14[9], Rab7[10], ectopic P-granules autophagy protein 5 homolog (EPG5)[11], the homotypic fusion and protein sorting (HOPS)-tethering complex[12], synaptosome-associated protein 29 (SNAP29), vesicle-associated membrane protein 8 (VAMP8), Syntaxin 17 (STX17)[13], and GABA type A receptor-associated proteins (GABARAPs)[14]. However, the molecular mechanisms regulating autophagosome-lysosome fusion are poorly understood.

Autophagy and the ubiquitin system are well-orchestrated to target cargo for degradation[15,16]. However, the interplay between these two pathways is not clear. Although poly-ubiquitinated proteins and damaged mitochondria and ubiquitin-coated bacteria in mammalian cells are well-defined cargoes for autolysosome-dependent degradation[15,16], the key ubiquitin enzymes, including E2-conjugating enzymes, E3 ligases, and deubiquitinases (DUBs), which regulate autophagy are not known. Furthermore, it is unclear whether nutrient starvation-induced autophagy (non-selective autophagy) is regulated by proteasomal-independent ubiquitin signaling.

To identify novel positive regulators of autophagy, we performed an RNA interference (RNAi) screen targeting 680 ubiquitin regulators and 30 well-known autophagy regulators as positive controls. We uncovered the "Baculovirus IAP Repeat (BIR) repeat-containing ubiquitin-conjugating enzyme" (BRUCE)/Apollon/Birc6 as the highest hit, along with known essential autophagy regulators. BRUCE is a member of the inhibitor of apoptosis (IAP) family and, as such, inhibits apoptosis by ubiquitinating apoptosis regulators such as Caspase 9 and Smac/Direct IAP-Binding Protein With Low PI (DIABLO), thereby targeting them for proteasomal degradation[17–20]. Here we establish BRUCE as a universal regulator of autophagy. We find that BRUCE regulates autophagosome–lysosome fusion independent from its ubiquitination activity. Our data reveal that BRUCE is a multifunctional regulator of apoptosis and autophagy.

## Results

### An RNAi screen identifies macroautophagy regulators.
To identify novel autophagy regulators, we performed an RNAi screen to identify new regulators of starvation-induced autophagy (Fig. 1a). We assessed autophagy within single cells by monitoring the fluorescence of an autophagy reporter, the mCherry-EGFP-tagged microtubule associated protein 1 light chain 3β (LC3B)[21] (Supplementary Fig. 1a). Unlike mCherry, EGFP is sensitive to low pH and thus the EGFP signal in mCherry-EGFP-LC3B-positive autophagosomes is quenched upon fusion with lysosomes[21,22]. Therefore, cells with intact autophagy have a low green fluorescent protein (GFP) signal, whereas those with defective autophagy have a high GFP signal (Fig. 1a).

We established monoclonal mouse embryonic fibroblast (MEF) reporter lines stably expressing mCherry-EGFP-LC3B (Supplementary Fig. 1b–e)[21,23]. We used clone #1 for our screen, as it showed the clearest reduction in flow cytometry-based GFP signal upon starvation (Supplementary Fig. 1c and d). Furthermore, the preferential loss of GFP signal induced by starvation of clone #1 was abrogated by lysosomal neutralization with the vacuolar type H+-ATPase (V-ATPase) inhibitor, Bafilomycin A1 (Supplementary Fig. 1e), confirming reporter function.

To systematically identify ubiquitin modifiers that regulate autophagy, we designed a miRNA-based short hairpin RNA (shRNA) (shRNAmir) retroviral library targeting all major E1 enzymes for ubiquitin and ubiquitin-like modifiers E2 enzymes and E3 ligases, and their complex components DUBs, other ubiquitin-related proteins, as well as 30 established autophagy regulators for positive controls (in total 710 genes/ 4,184 shRNAmirs; Supplementary Data 1). We introduced one shRNA/reporter cell by transduction, starved the cells, and sorted them by fluorescence-activated cell sorting (FACS) into populations of high GFP signal (defective in autophagy) or low GFP signal (functional autophagy) (Fig. 1a, b and Supplementary Fig. 2a, b and c). Knockdown of ATG5, which is required for autolysosome formation, led to an increased GFP signal in clone #1 cells, as expected (Fig. 1b). We analyzed the shRNAmirs enriched in the high-GFP gate by next-generation sequencing (NGS) and identified nine genes as positive autophagy regulators. Importantly, we uncovered major regulators of autophagy, including five Atg genes, which validates our screen (Table 1).

We verified that Tsg101[24], Sqstm1/p62, and Nedd8[25], which were previously implicated in autophagy regulation, as well as a novel hit, Bruce, regulate autophagic flux. We generated stable shRNA reporter-MEFs lines using two distinct shRNAs per gene. Indeed, reduced expression of these genes led to defects in autophagy (Fig. 1c–e and Supplementary Fig. 2d–f). Similarly, Bruce−/− MEFs displayed defective autophagy compared to wild-type (WT) MEFs (Fig. 2a). Upon starvation, WT MEFs expressing mCherry-EGFP-LC3B accumulated foci that were positive for mCherry but negative for GFP, consistent with the formation of autolysosomes and intact autophagic flux. In contrast, most of the autophagosomes induced by starvation in Bruce−/− MEFs expressing the mCherry-EGFP-LC3B reporter remained mCherry-GFP double positive (Fig. 2b, c), suggesting that autophagic flux is inhibited by BRUCE deficiency. The effect of BRUCE deficiency on autophagic flux was not limited to starvation-induced autophagy, but also observed in mitophagy induced by Antimycin A and Oligomycin (Supplementary Fig. 3a). Collectively, our findings establish that BRUCE positively regulates autophagic flux.

**BRUCE regulates autophagic flux**. Next, we evaluated the levels of autophagosome markers, including the ATG8 family members LC3B, GABARAP, GABARAPL1, as well as the autophagy substrate p62. The ATG8 family members are known to be lipidated upon autophagosome formation and degraded at a later step inside autolysosomes, similar to p62. As expected, starvation reduced the protein levels of these markers in WT MEFs (Fig. 2d and Supplementary Fig. 3b–e), whereas Bafilomycin A1 treatment inhibited the degradation of LC3B, especially the lipidated form LC3B-II (Fig. 2d and Supplementary Fig. 3b–e). Bruce−/− MEFs displayed elevated protein levels of LC3B-I, LC3B-II, GABARAP, and GABARAPL1 relative to WT MEFs, in both the control and starved conditions (Fig. 2d). Other cell lines depleted for BRUCE, including two shBruce MEF lines, two CRISPR-gBruce MEF lines, and two CRISPR-gBruce HAP1 cells (a near-haploid human

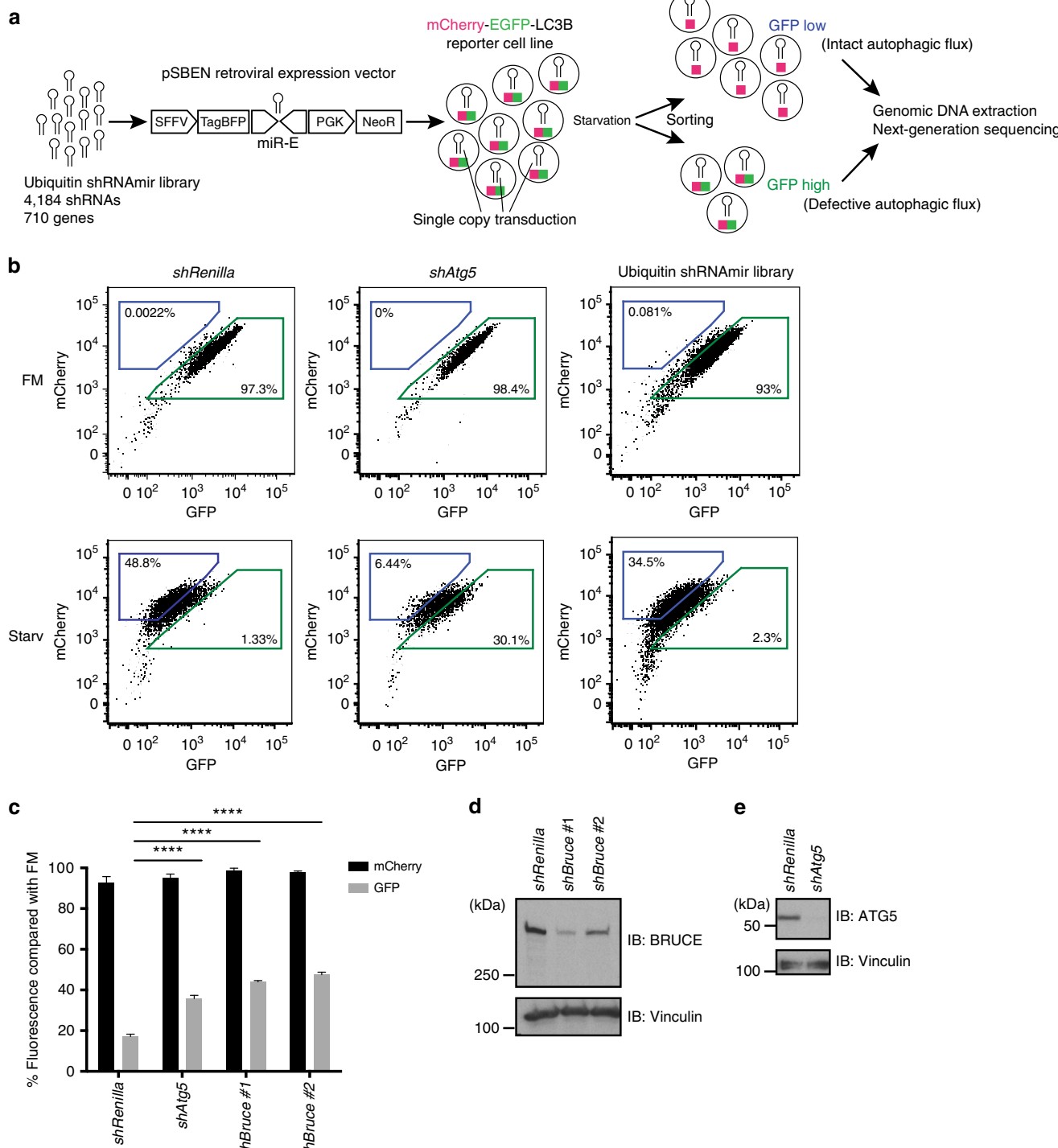

**Fig. 1** shRNA-based screen identifies BRUCE as an autophagy regulator. **a** Schematic representation of the FACS-based screen using a library of short hairpin RNAs with the miR-E structure (shRNAmir). shRNAs (4,184) targeting 710 ubiquitin- and autophagy-associated genes were cloned into a retroviral gene expression vector (pSBEN). A single copy of shRNA was delivered into the mCherry-EGFP-LC3B-expressing MEF line (clone 1) by transduction. A cell population selected with neomycin was starved and sorted based on the GFP fluorescent signal. Subsequently, genomic DNA was isolated from sorted cells and analyzed by next-generation sequencing. miR-E, microRNA-element 3′ backbone; NeoR, Neo reporter gene cassette; PGK, phosphoglycerate kinase; SFFV, spleen focus-forming virus; TagBFP, monomeric blue fluorescent protein. **b** Gating strategy of "GFP low/high" cell population of *shRenilla* and *shAtg5* MEFs, compared with ubiquitin shRNAmir library transduced cells (replicate #1). After 6 h starvation, cells were enriched in "GFP low" gate (blue) and depleted in "GFP high" gate (green). Percentage of cells in corresponding gates is indicated. **c** Relative mCherry and GFP signals compared with basal condition (fully supplemented medium (FM)) analyzed by flow cytometry. Control (*shRenilla* (Renilla.713)), two BRUCE knockdown (KD) (*shBruce*#1 (Birc6.766.), and #2 (Birc6.14744)) and ATG5 KD (*shAtg5* (Atg5.1063)) MEF lines were analyzed. Data are presented as mean±SD from three biological replicates (****$p < 0.0001$). Representative data are shown from four independent experiments. **d**, **e** Knockdown efficiency of BRUCE and ATG5 in *shBruce*#1 and #2, and *shAtg5* MEFs, respectively, analyzed by immunoblotting using anti-BRUCE and anti-ATG5 antibodies. Anti-Vinculin antibody was used for loading control

**Table 1 shRNA-based screen identifies nine positive autophagy regulators using restrictive analysis criteria**

| Gene | # Scoring shRNAs | Avg Geomean |
|---|---|---|
| *Atg7* | **5** | **21.60** |
| *Atg16l1* | **4** | **7.59** |
| Birc6 | 4 | 3.18 |
| *Rb1cc1* | **3** | **14.33** |
| Tsg101 | 3 | 7.15 |
| *Atg12* | **3** | **4.91** |
| Sqstm1 | 3 | 3.77 |
| Nedd8 | 3 | 3.05 |
| *Atg9a* | **3** | **3.03** |

Hit list showing genes with highest scoring shRNAs, sorted based on the number of scoring shRNAs (# scoring shRNAs) and the average geometric mean (Avg Geomean) of the fold enrichment of shRNAs in the "GFP high" gate, compared with the "GFP low" gate. Established autophagy regulators are shown in bold

cells), displayed similar results for LC3B (Supplementary Fig. 3d–f). In addition, we evaluated BRUCE-deficient haploid mouse embryonic stem cell (mESC) clones and sister clones with restored BRUCE expression (Fig. 2e). $Bruce^{-/-}$ mESCs showed elevated LC3B and GABARAP in control and starved conditions compared with both the rescued sister clones and parental WT mESCs (Fig. 2f and Supplementary Fig. 3g). In addition, the level of the autophagy substrate p62 was increased in the basal condition in shBruce MEFs and in positive control shAtg5 MEFs with disrupted autolysosome formation, relative to control MEFs (Supplementary Fig. 3b and h). In contrast, the transcript levels of LC3B, GABARAP, and GABARAPL1, as well as p62, were not largely affected by BRUCE depletion in both basal and starved conditions, as revealed by RNA sequencing (RNA-Seq) (Supplementary Fig. 3i). Interestingly, the total ubiquitin signal was clearly increased in shBruce MEFs compared with control MEFs in both basal and starved conditions (Supplementary Fig 3b). Together, these data suggest that BRUCE is involved in a general autophagic process.

**BRUCE selectively interacts with GABARAP and GABARAPL1**. Next, we examined whether BRUCE might regulate autophagic flux as an autophagy receptor, which links autophagosome membranes and autophagy targets by interacting with ATG8 family members and ubiquitin, respectively. We performed pull-down assays using glutathione S-transferase (GST)-tagged recombinant ATG8 family members or ubiquitins of different lengths immobilized with agarose beads, and total cell extracts of HEK293T cells transiently expressing full-length or various mutants of Myc-tagged BRUCE (Fig. 3a, b and Supplementary Fig. 4a–h). We found that full-length BRUCE interacted with GABARAP and GABARAPL1, but not with other ATG8 family members, LC3A, LC3B, LC3C, or GABARAPL2, or with ubiquitins (Fig. 3a). A catalytically inactive point mutant of BRUCE (C4638A) interacted with GABARAP and GABARAPL1, similar to WT BRUCE (Supplementary Fig. 4a). All the BRUCE deletion fragments interacted with GABARAP and GABARAPL1, except for aa 1–1360, aa 259–332 (the BIR domain), and aa 4569–4707 (the C-terminal Ubiquitin-conjugating (UBC) domain) (Supplementary Fig. 4b–h). p62 bound all the ATG8 family members, as expected (Supplementary Fig. 4i and j). Most ATG8-binding partners, such as p62, interact with ATG8 via an LC3-interacting region (LIR)[26–28]. GABARAP mutants with a mutated LIR-binding surface did not interact with p62, as expected (Supplementary Fig. 4k); however, they retained the interaction with BRUCE (Fig. 3c and Supplementary

Fig. 4l). These data suggest that BRUCE does not use a canonical LIR-motif to interact with GABARAP. Collectively, our data propose that BRUCE specifically interacts with the ATG8 family members GABARAP and GABARAPL1 via multiple binding regions that are outside of the BIR and the UBC domains (Supplementary Fig. 4m), and are not canonical LIR-containing regions.

As BRUCE interacted preferentially with GABARAP and GABARAPL1, we asked whether starvation-dependent lysosome targeting of GABARAP and GABARAPL1 is affected by BRUCE depletion using mCherry-EGFP-tagged reporters. Similar to mCherry-EGFP-LC3B, mCherry-EGFP-GABARAP and -GABARAPL1 pH sensors monitor autolysosome formation upon starvation in WT and control (shRenilla) MEFs; starvation leads to a significant decrease of GFP signal compared with the basal condition (Fig. 3d, e). Knockdown of BRUCE and of ATG5 significantly suppressed the loss of GFP signal in MEFs expressing mCherry-EGFP-GABARAP and mCherry-EGFP-GABARAPL1 (Fig. 3d, e). These findings are consistent with previous observations in MEFs expressing mCherry-EGFP-LC3B (Fig. 1c). Interestingly, the protein level of BRUCE itself was not affected by starvation (Fig. 2d), or by ATG5 deficiency (Supplementary Fig. 4n and o), suggesting that BRUCE is not a typical autophagy substrate like p62, which was stabilized in ATG5 deficient MEFs (Supplementary Fig. 3h and 4n). Together, our findings show that BRUCE interacts with GABARAP and GABARAPL1, which are suggested to regulate a later step of the autophagy pathway. Therefore, BRUCE might have a role in the maturation step of the autophagy pathway.

**BRUCE does not regulate lysosomal functions**. Our results suggest that BRUCE acts at a late stage in the autophagy pathway; therefore, we analyzed whether BRUCE colocalizes with lysosomes. In WT MEFs, endogenous BRUCE localized closely to lysosome-associated membrane protein 2 (LAMP2) in control and starved conditions with or without Bafilomycin A1 treatment (Fig. 4a and Supplementary Fig. 5a). The colocalization signal of BRUCE and LAMP2 was enhanced in starvation condition with Bafilomycin A1 treatment, suggesting that BRUCE prefers autolysosomes over lysosomes. However, colocalization of BRUCE and LAMP2 was partially and not entirely dependent on ATG5, suggesting that BRUCE localizes to both lysosomes and autolysosomes (Fig. 4a). We next analyzed lysosomal pH, structure, and biogenesis in BRUCE-deficient cells upon starvation by monitoring Lysotracker Red, which labels acidic organelles. We did not observe significant differences between control and $Bruce^{-/-}$ or gBruce MEFs under the control or the starved conditions by FACS (Fig. 4b and Supplementary Fig. 5b). Similar results were observed for shAtg5 MEFs, as expected (Supplementary Fig. 5c). The mRNA levels of transcription factor EB (TFEB) target genes, which are critical for lysosomal biogenesis[29], were also not differentially expressed in shBruce MEFs compared with control MEFs (Supplementary Fig. 5d). These findings suggest that BRUCE localizes to lysosomes but is not involved in regulating lysosome biogenesis nor lysosome functions.

**BRUCE promotes autophagosome–lysosome fusion**. As BRUCE depletion affected autophagic flux (Fig. 1c and 2a) and colocalized with LAMP2 (Fig. 4a), we next investigated whether BRUCE regulates fusion of the autophagosome and lysosome. We analyzed colocalization of LC3B and LAMP2, which associate with the autophagosome and lysosome, respectively. Confocal microscopy of WT MEFs revealed many LC3B-positive vesicles within LAMP2-positive lysosomal structures, especially after starvation, which became most apparent when cells were

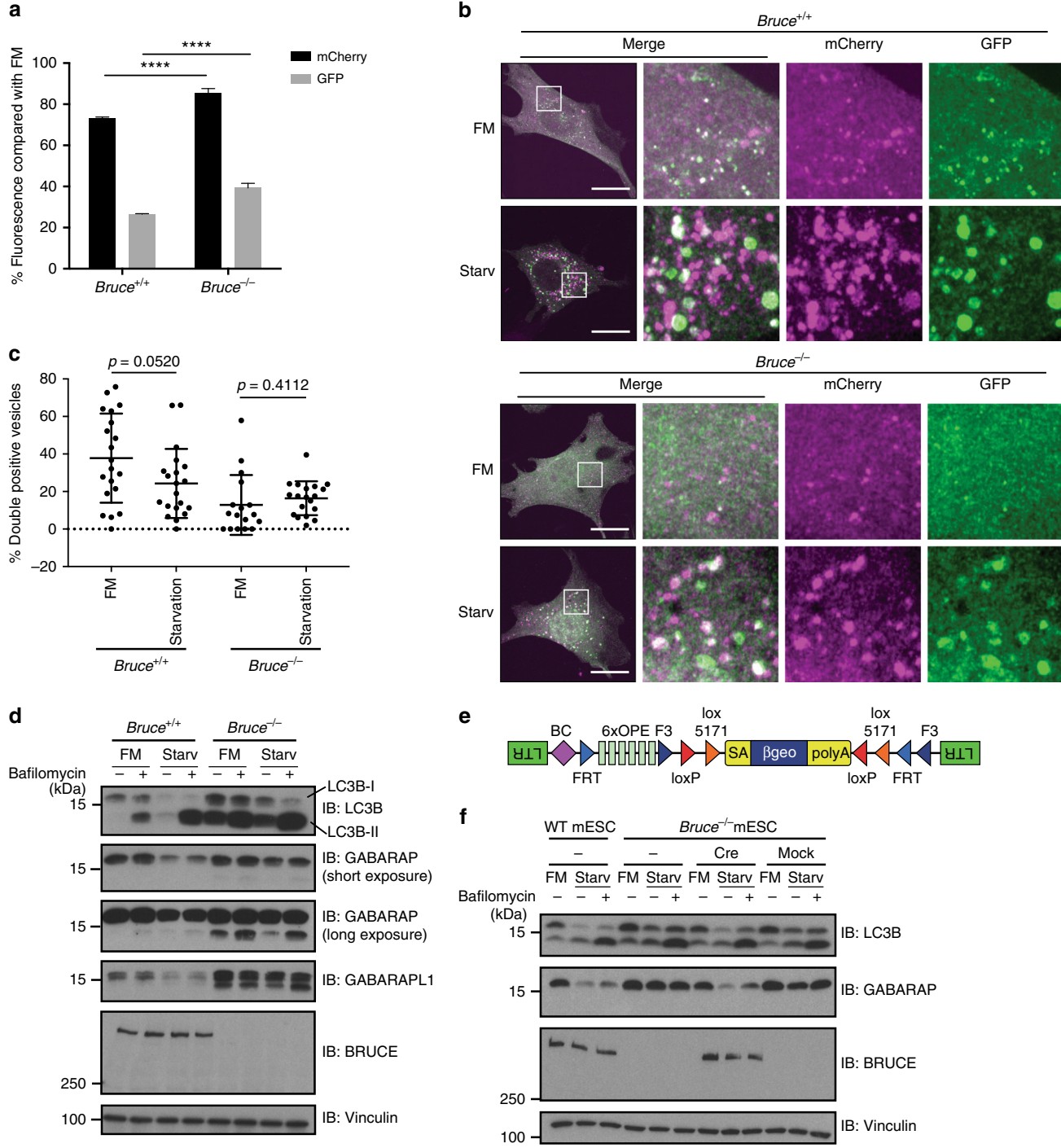

**Fig. 2** BRUCE is required for starvation-induced degradation of ATG8 family proteins. **a** Normalized mCherry and GFP signals after starvation compared to basal condition (FM) analyzed by flow cytometry in *Bruce*[+/+] and *Bruce*[−/−] MEFs stably expressing mCherry-EGFP-LC3B. Data are presented as mean±SD from three biological replicates (****$p < 0.0001$). Representative data are shown from four independent experiments. **b** Confocal microscopy images of *Bruce*[+/+] and *Bruce*[−/−] MEFs stably expressing mCherry-EGFP-LC3B in different conditions. FM, fully supplemented medium; Starv, starvation medium for 4 h. Scale bars, 20 μm. **c** Quantification of mCherry-GFP double positive spots in *Bruce*[+/+] and *Bruce*[−/−] MEFs in regular and starvation conditions (as in **b**). Data are presented as dot plots with mean±SD. $n = 20$ cells. *p*-values are indicated and analyzed by *t*-test. **d** Protein levels of autophagy markers in *Bruce*[+/+] and *Bruce*[−/−] MEFs under basal (FM) and 2 h-starved (Starv) conditions with or without Bafilomycin A1. **e** A schematic of the inserted gene trap cassette in *Bruce*[−/−] haploid mouse embryonic stem cells (mESCs). **f** Protein levels of autophagy markers in *Bruce*[+/+] and *Bruce*[−/−] clone #1 haploid mESCs under basal (FM) and 2 h-starved (Starv) conditions with or without Bafilomycin A1. BRUCE expression was recovered by Cre recombinase in *Bruce*[−/−] mESCs and compared with mock infected mESCs

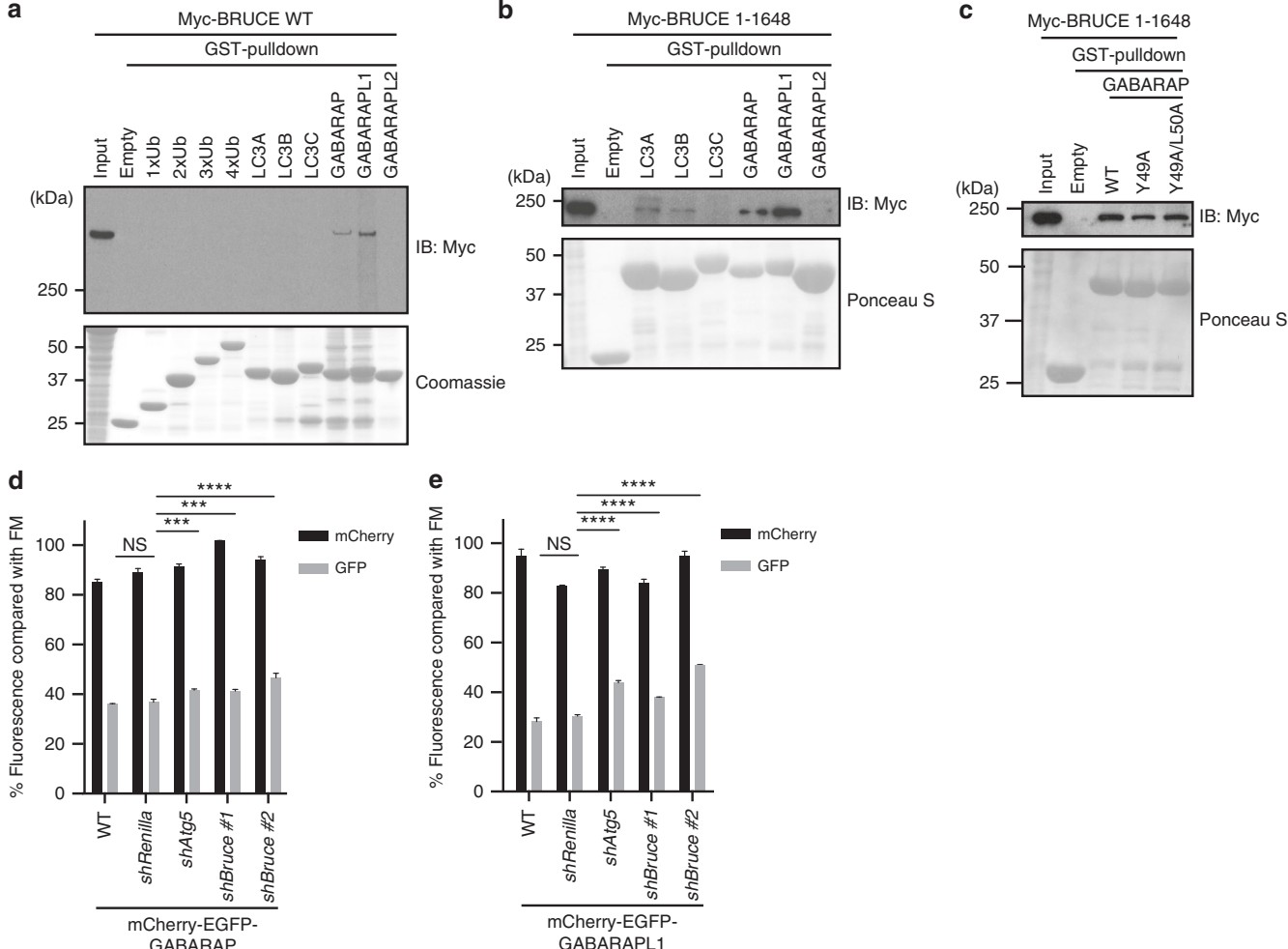

**Fig. 3** BRUCE interacts specifically with GABARAP and GABARAPL1. **a**, **b** Interaction of BRUCE wild type (WT) or aa 1–1648 fragment and mammalian ATG8 proteins and ubiquitins examined by GST pull-down assay. Total cell lysate of HEK293T cells transfected with Myc-tagged WT or the 1–1648 fragment was incubated with GST-ubiquitin or GST-ATG8 proteins immobilized with glutathione sepharose beads. Interaction was detected by immunoblotting using anti-Myc antibody. The amount of GST-protein used was compared by Coomassie or Ponceau S staining. **c** Interaction of BRUCE 1–1648 and GABARAP WT or mutants examined by GST pull-down assay. GABARAP mutants contained substitutions in the canonical LC3-interacting region (LIR) recognition surface (Y49A and Y49A/L50A). **d**, **e** Normalized mCherry/GFP signals under starvation condition normalized to FM condition in MEF lines stably expressing mCherry-EGFP-GABARAP **d** or mCherry-EGFP-GABARAPL1 **e**. Control (WT and *shRenilla*) and KD (*shAtg5* and *shBruce#1* and *#2*) cells were starved for 6 h. Data are presented as mean±SD from three biological replicates (****$p < 0.0001$, ***$p < 0.001$, NS, not significant). Representative data are shown from three independent experiments

additionally treated with Bafilomycin A1 (Fig. 5a, b). Compared with WT MEFs, *Bruce*$^{-/-}$ MEFs displayed a substantially reduced number of LC3B-containing LAMP2-positive lysosomal structures (Fig. 5a, b). As a control, we examined LC3B and LAMP2 signals in ATG5-deficient MEFs and did not observe LC3B-containing LAMP2 structures, as expected (Supplementary Fig. 6a). As BRUCE interacts with GABARAP (Fig. 3a), we examined whether the fusion of GABARAP-positive autophagosomes and lysosomes also requires BRUCE. As observed for the localization of endogenous LC3B and LAMP2 in MEFs, mCherry-GABARAP was observed within LAMP1-mGFP-positive structures in control HAP1 cells, whereas they largely remain outside of the LAMP1-positive structures in CRISPR-*gBRUCE* HAP1 cells (Supplementary Fig. 6b). These observations were supported by electron microscopy, showing that *Bruce*$^{-/-}$ MEFs accumulated autophagosomes upon starvation compared with WT MEFs (Fig. 5c). These data strongly suggest that BRUCE regulates the fusion of autophagosomes and lysosomes.

Importantly, the endocytosis-dependent lysosomal pathway was not affected in *Bruce*$^{-/-}$ MEFs, as determined by treatment with DQ-Ovalbumin, a self-quenched conjugate of ovalbumin that exhibits bright green fluorescence upon proteolytic degradation (Fig. 5d), and by degradation of epidermal growth factor receptor (EGFR) (Supplementary Fig. 6c). These data indicate that BRUCE specifically affects the formation of autolysosomes, but not the endosomal pathway. Interestingly, the basal expression level of EGFR in *Bruce*$^{-/-}$ MEFs was clearly lower compared to WT MEFs (Supplementary Fig. 6c and d). The protein level of EGFR was not affected by Bafilomycin A1 treatment in WT and *Bruce*$^{-/-}$ MEFs cells (Supplementary Fig. 6d), suggesting that the low level of EGFR in *Bruce*$^{-/-}$ MEFs is independent of the lysosome.

**BRUCE regulates cellular localization of STX17.** To further understand how BRUCE regulates autophagosome-lysosome

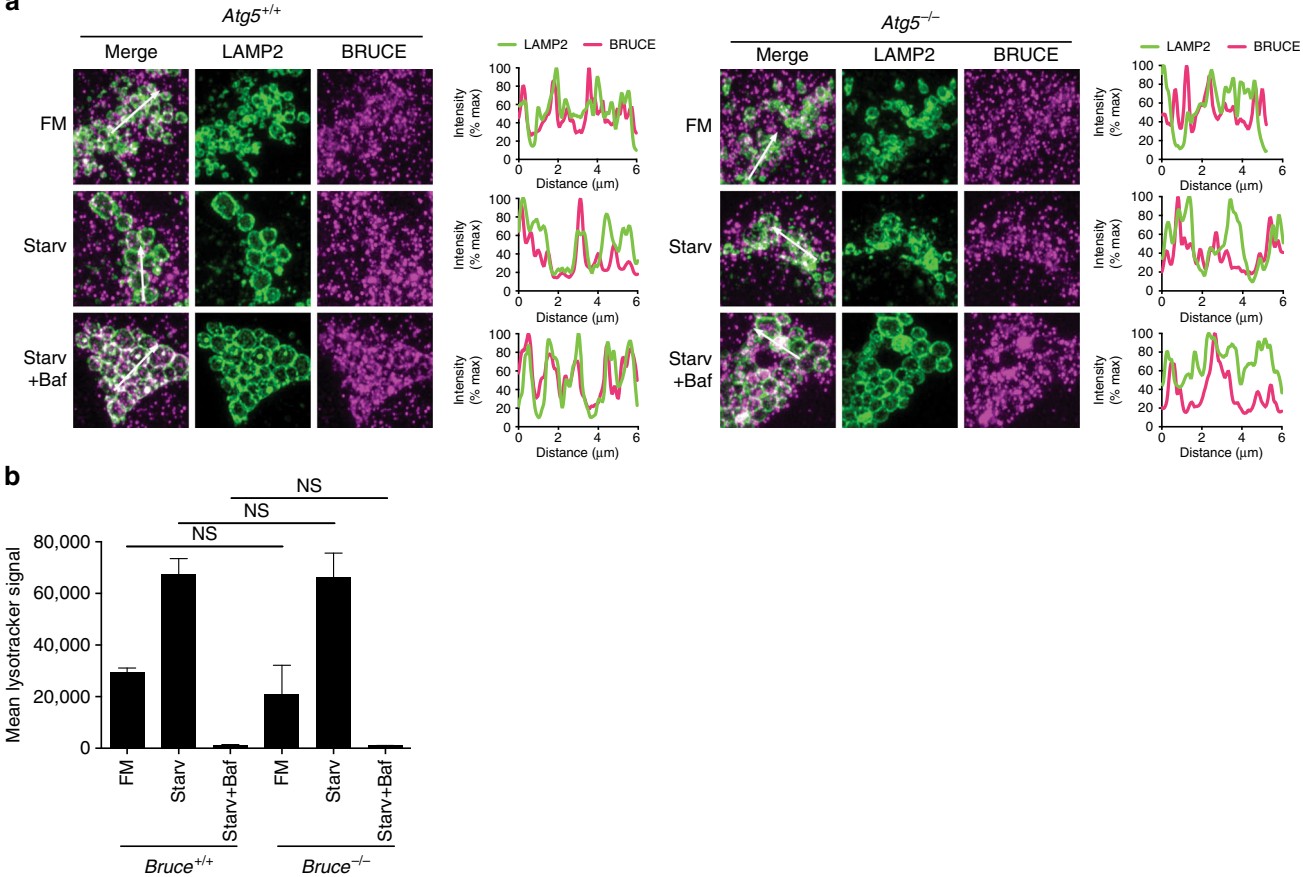

**Fig. 4** BRUCE localizes to lysosomes but has no effect on lysosomal functions. **a** Immunofluorescent staining of endogenous LAMP2 and BRUCE in $Atg5^{+/+}$ and $Atg5^{-/-}$ MEFs. $Atg5^{+/+}$ and $Atg5^{-/-}$ MEFs in FM, starved for 4 h with or without Bafilomycin A1 (100 nM) were fixed and stained with anti- LAMP2 and anti- BRUCE antibodies. Fluorescent intensity of LAMP2 and BRUCE signals across 6 μm regions marked with arrows is shown in line plots. **b** Mean LysoTracker Red (LTR) signal in $Bruce^{+/+}$ and $Bruce^{-/-}$ MEFs cells analyzed by flow cytometry. MEFs grown in regular and 6 h-starved conditions with or without Bafilomycin A1. Data are presented as mean±SD from three biological replicates (NS, not significant). Representative data are shown from three independent experiments

fusion, we investigated whether BRUCE interacts with known regulators of autophagosome-lysosome fusion (Fig. 6a and Supplementary Fig. 7a–i). We evaluated components of the core fusion machinery, including Rab7, two HOPS complex components VPS33A and VPS16, STX17, SNAP29, VAMP8, ATG14, Pleckstrin homology domain-containing family M member 1 (PLEKHM1), as well as lysosomal membrane proteins LAMP1 and LAMP2. We found that transiently expressed GFP-STX17 and GFP-SNAP29 co-immunoprecipitated with Myc-BRUCE from HEK293T cells (Fig. 6a and Supplementary Fig. 7a), suggesting that BRUCE interacts with the autophagosomal soluble NSF attachment protein receptor (SNARE) complex consisting of STX17 and SNAP29, but not with VAMP8, the proposed lysosomal SNARE.

STX17 is a critical regulator of autophagosome-lysosome fusion and is specifically localized at mature autophagosomes. BRUCE interacts with STX17, so we investigated whether BRUCE regulates the cellular localization of STX17 (Fig. 6b and Supplementary Movie 1 and 2). In $Bruce^{+/+}$ MEFs stably expressing GFP-STX17, GFP-STX17-positive vesicles were formed upon amino acid starvation as expected (Fig. 6b). In contrast, some STX17-positive vesicles accumulated in $Bruce^{-/-}$ MEFs even under the basal condition, which were further induced upon starvation (Fig. 6b and Supplementary Movie 1). We found that GFP-STX17 colocalization with LC3B-positive vesicles was

enhanced in $Bruce^{-/-}$ MEFs compared with $Bruce^{+/+}$ MEFs (Fig. 6c), suggesting that the dynamic dissociation of STX17 from the autophagosome is disrupted in BRUCE-deficient cells. These data suggest that BRUCE regulates STX17-mediated fusion of autophagosomes and lysosomes.

**A non-catalytic BRUCE fragment restores autophagic flux.** To further elucidate how BRUCE regulates autophagy, we determined if starvation-induced autophagy in $Bruce^{-/-}$ MEFs is rescued by the non-catalytic BIR-containing fragment (aa 1–1648), which is the minimum N-terminal fragment interacting with GABARAP and GABARAPL1 (Fig. 3b and Supplementary Fig. 4m). The protein levels of LC3B, GABARAP, GABARAPL1, and p62 in $Bruce^{-/-}$ MEFs were partially rescued by expression of the BRUCE (aa 1–1648) fragment (Fig. 7a). Importantly, the expression of the BRUCE (aa 1–1648) fragment in $Bruce^{-/-}$ MEFs also partially rescued the formation of LC3B-containing LAMP2-positive structures (Fig. 7b, c). Similar to full-length BRUCE, BRUCE (aa 1–1648) also colocalized with LAMP2 in reconstituted cells under basal and starved conditions (Fig. 7d). Further, GFP-STX17 co-immunoprecipitated with the Myc-tagged BRUCE (aa 1–1648) fragment (Fig. 7e), suggesting that this region is sufficient for the interaction with STX17. Collectively, these results suggest

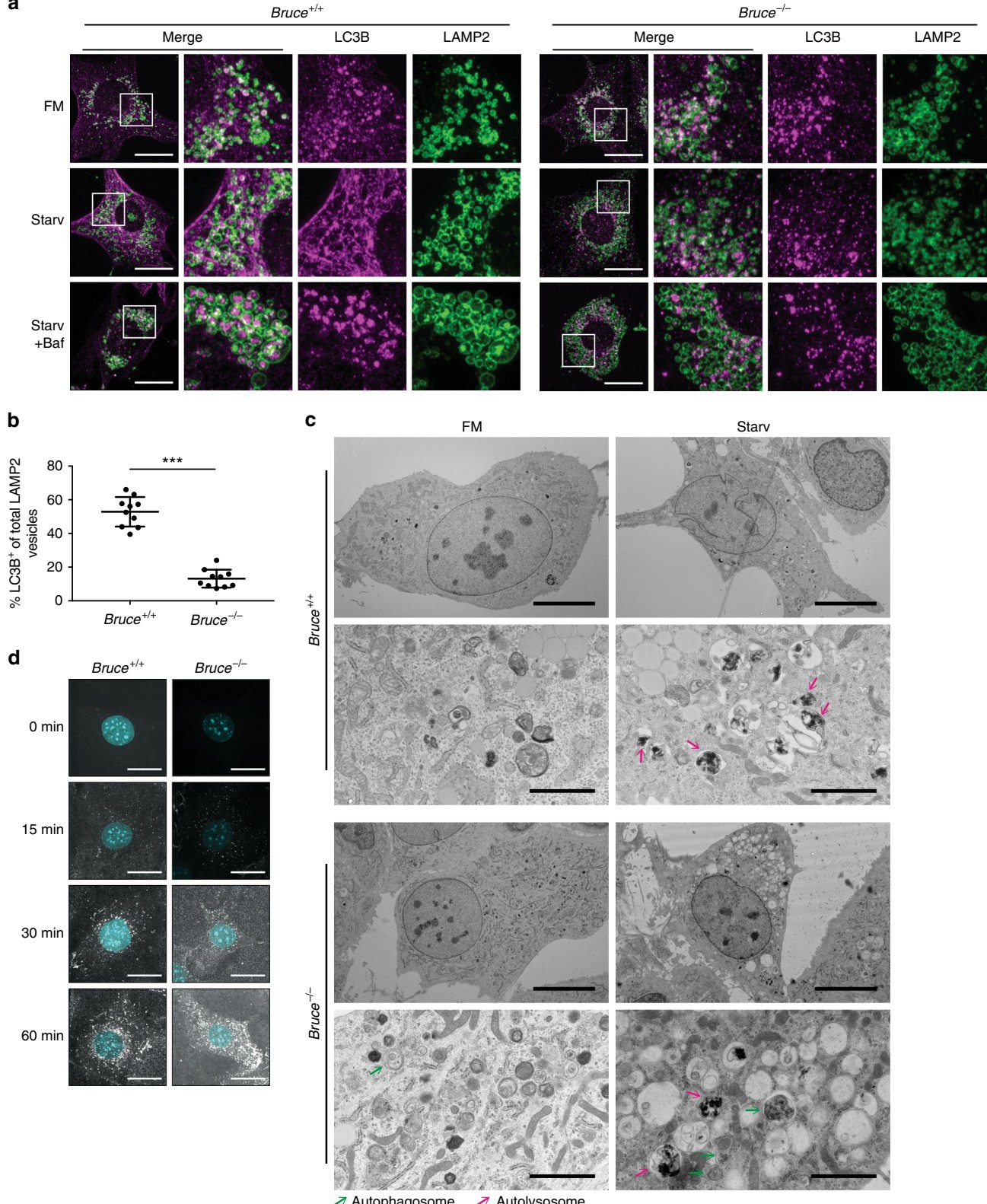

**Fig. 5** BRUCE controls autolysosome formation. **a** Immunofluorescent staining of endogenous LC3B and LAMP2 in *Bruce*[+/+] and *Bruce*[−/−] MEFs. *Bruce*[+/+] and *Bruce*[−/−] MEFs in fully supplemented medium (FM), starved (Starv) for 4 h with or without Bafilomycin A1 (Baf; 100 nM) were fixed and stained with anti-BRUCE and anti-LAMP2 antibodies. Scale bars, 20 μm. **b** Percentage of LAMP2-positive vesicles containing LC3B-positive aggregates in MEFs under starvation condition with Bafilomycin A1 treatment as in **a**. Data are presented as dot plots with mean±SD (***$p < 0.001$, analyzed by *t*-test); $n = 10$ cells. **c** Electron microscopic images of *Bruce*[+/+] and *Bruce*[−/−] MEFs in FM or starved for 2 h. Overview: scale bars, 10 μm; magnified: scale bars, 2 μm. Green and pink arrows point to autophagosomes and autolysosomes, respectively. **d** Examination of the endosomal/lysosomal pathway by confocal microscopy using DQ-Ovalbumin in *Bruce*[+/+] and *Bruce*[−/−] MEFs. Cells were grown in DQ-Ovalbumin containing starvation media for the indicated time. Scale bars, 20 μm

that BRUCE regulates mammalian autophagy as a linker protein between the autophagosome and lysosome (Fig. 8), and reveal a unique non-catalytic function of BRUCE distinct from its role in apoptosis.

## Discussion

In this study, we identified the IAP family member BRUCE[19,20] as a new autophagy regulator. BRUCE inhibits apoptosis by ubiquitinating and promoting degradation of apoptosis

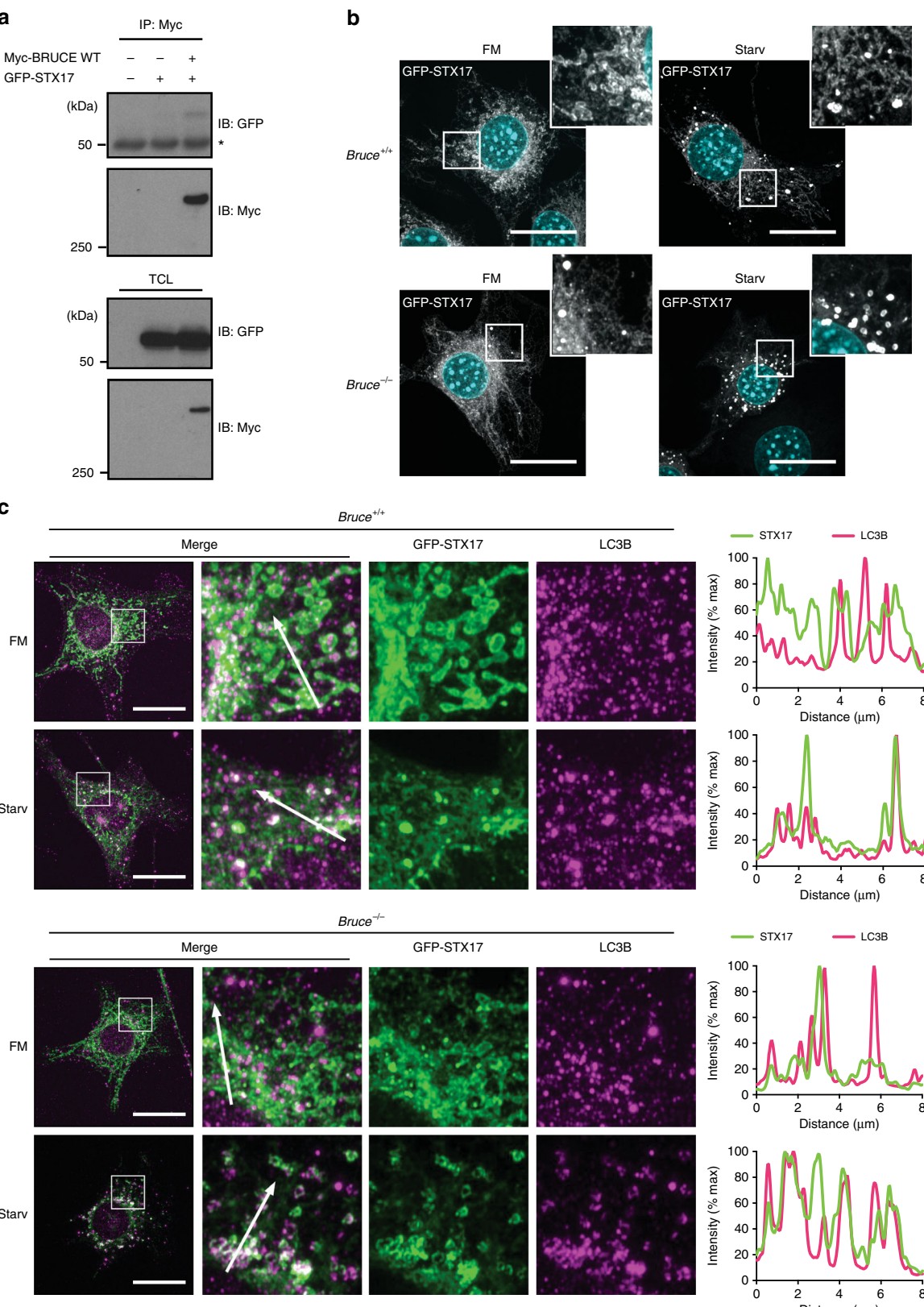

regulators[19,30]. Here we found that autophagic flux is inhibited by BRUCE depletion, whereas the initiation of autophagosome formation is largely unaffected. BRUCE interacts preferentially with GABARAP and GABARAPL1 among six ATG8 family members, via multiple binding regions within BRUCE. Our results indicate that BRUCE regulates GABARAP- (or GABARAPL1-) positive autophagosome fusion with the lysosome. As ATG8 family members are structurally similar and share many interacting partners[31,32], whether GABARAP proteins have distinct functions in autophagy from the other ATG8 members is not yet well understood. Our data suggest that BRUCE interacts with GABARAP and GABARAPL1 via non-canonical protein surfaces, based on pulldown assays using GABARAP mutants; however, further studies are required to understand this at the structural level. BRUCE protein levels do not decrease upon starvation, indicating that BRUCE is not a typical autophagy substrate such as p62, Next to BRCA1 gene 1 protein (NBR1), Optineurin or Calcium-binding and coiled-coil domain-containing protein 2 (CALCOCO2)/NDP52[33]. The role of BRUCE seems similar to that of PLEKHM1, which links ATG8 and the HOPS complex, regulating autolysosome formation[34]. However, unlike BRUCE, PLEKHM1 interacts with all ATG8 members and regulates the endocytosis-dependent EGFR degradation pathway. Importantly, we found that BRUCE interacts with STX17 and SNAP29, which are critical regulators of autophagosome–lysosome fusion. Furthermore, BRUCE deficiency in cells does not inhibit the formation of STX17-positive autophagosomes; indeed, STX17-positive vesicles are accumulated in BRUCE-deficient cells even in the absence of starvation. These data indicate that mature autophagosomes form in BRUCE-deficient cells, and that turnover of STX17-positive autophagosomes is inhibited, consistent with a deficiency of autophagosome–lysosome fusion.

It has been shown that mammalian target of rapamycin (mTOR) has a critical role in reformation of lysosomes[7,35], whereas the Ras-related proteins (RAB) and the SNARE[7] function in trafficking and vesicle fusion. The mRNA levels of mTOR, RABs, and SNAREs, as well as lysosomal biogenesis-related genes including the TFEB targets[29,36] were not affected by BRUCE knockdown.

Autophagy has been shown to be critical for tumor progression in different contexts[37,38]. Previous studies have implicated BRUCE in cancer[39–43], although its roles are poorly understood compared with other IAP family members. We propose that BRUCE may influence tumor progression not only via its anti-apoptotic function but also via its regulation of autophagy. Further studies are required to understand the role of BRUCE in the regulation of mammalian autophagy in vivo and its potential role in cancer.

## Methods

**Plasmids**. pcDNA3-Myc-BRUCE FL (aa 1–4829) and mutants (aa 1–1648, aa 1–1360, and C4638A) were a kind gift from Mikihiko Naito[19]. Additional fragments in pcDNA3 used in this study were subcloned by Gibson Assembly using EcoRI/XhoI restriction sites. pBabe-puro-BRUCE aa 1–1648 was generated using a standard subcloning method. GST-tagged human MAP1LC3A, human

MAP1LC3B, human MAP1LC3C, human GABARAP, mouse GABARAPL1, and human GABARAPL2 in pETM30 vector were kindly provided by Felix Randow[44]. GST-GABARAP LIR docking site mutants were generated using a standard site-directed mutagenesis method. pmCherryC1-GABARAP, pmCherryC1-GABARAPL1, pBabe-puro-mCherry-EGFP-GABARAP, and pBABE-puro-mCherry-EGFP-GABARAPL1 were generated using a standard subcloning method. pCMV-Gag-Pol was used as a helper plasmid for retrovirus production, and shRNAs were cloned into pRSF91-SFFV-TagBFP-mirE-PGK-Neo-WPRE as previously described[45]. pGex4T1-human ubiquitin 1×, 2×, 3×, and 4× were described elsewhere[46]. pBabe-puro mCherry-EGFP-LC3B was a gift from Jayanta Debnath (Addgene plasmid #22418)[21], LAMP1-mGFP was a gift from Esteban Dell'Angelica (Addgene plasmid #34831)[47], pMRXIP GFP-STX17 WT (Addgene plasmid #45909)[13], pMRXIP GFP-SNAP29 (Addgene plasmid #45923)[13], pMRXIP GFP-VAMP8 (Addgene plasmid #45919)[13], pMXs-IP GFP-Atg14 (Addgene plasmid #38264)[48], pMXsIP GFP-VPS33A (Addgene plasmid #67022)[12], pMRXIP VPS16-GFP (Addgene plasmid #67023)[12] were gifts from Noboru Mizushima, h-Plekhm1-EGFP was a gift from Paul Odgren (Addgene plasmid #73836), GFP-Rab7 WT was a gift from Richard Pagano (Addgene plasmid #12605)[49], and HA-LAMP2 was a gift from Ana Maria Cuervo[50].

**Antibodies**. Anti-BRUCE (BD, 611193; 1:500), anti-Ubiquitin (P4D1, Santa Cruz, sc-8017; 1:1,000), anti-Myc (9E10, Covance, MMS-150P; 1:1,000), anti-GFP (Santa Cruz, sc-9996; 1:1,000), anti-mCherry (Clontech, 632543; 1:1,000), anti-Alpha-Tubulin (Abcam, ab15246; 1:1,000), anti-Vinculin (Sigma-Aldrich, V9131; 1:1,000), anti-ATG5 (Cell signaling, 8540; 1:1,000), anti-LC3 (Nano Tools, 0260-100/LC3-2G6; 1:100), anti-GABARAP (E1J4E, Cell Signaling, 13733; 1:1,000), anti-GABARAPL1 (Abcam, ab86497; 1:500), anti-P62/SQSTM1 (MBL, PM045; 1:1,000), anti-LAMP1 (Abcam, ab24170; 1:1,000), anti-LAMP2 (Abcam, ab13524; 1:500), anti-ATG4B (Cell signaling, 5299; 1:1,000), anti-ULK1 (Santa Cruz, sc-33182; 1:500), anti-Beclin-1 (Cell Signaling, 3738; 1:1,000) and anti-EGFR (D38B1, Cell Signaling, 4267; 1:1,000) antibodies were purchased and used according to the manufacturer's recommendations.

**Cell lines**. Bruce[+/+], Bruce[−/−] MEFs (a kind gift from Mikihiko Naito)[19], Atg5[+/+] and Atg5[−/−] MEFs[34,51], human embryonic kidney (HEK) 293 T cells (ATCC), and packaging cell lines, Platinum-E (Plat-E) (Ecotropic) and Platinum-A (Plat-A) (Amphotropic)[45,52,53] were maintained at 37 °C in 5% $CO_2$, in Dulbecco's modified Eagle medium high glucose (Sigma) supplemented with 10% fetal calf serum (Thermo Fisher Scientific), 100 U ml$^{-1}$ penicillin-streptomycin (Sigma), and 2 mM L-glutamine (Sigma-Aldrich). HAP1 cells were purchased from Haplogen/Horizon Genomics and were maintained in Iscove's modified Dulbecco's medium (Life technologies), supplemented with 10% fetal calf serum and 100 U ml$^{-1}$ penicillin–streptomycin. Mouse haploid ES cells[54] were a kind gift from Haplobank and were maintained in Dulbecco's modified Eagle medium high glucose supplemented with 13.6% fetal calf serum (Thermo Fisher Scientific), 100 U ml$^{-1}$ penicillin–streptomycin (Sigma), 2 mM L-glutamine (Sigma-Aldrich), 1 mM sodium pyruvate (Sigma), 1 × MEM non-essential amino acid solution (Sigma), 0.00035% β-mercaptoethanol, and ESGRO (Millipore). All the cell lines used in this study were mycoplasma negative.

**CRISPR/Cas9 BRUCE mutant cell lines**. Design of small guide RNAs (sgRNA) was performed using the online CRISPR Design Tool from Zhang Lab (http://crispr.mit.edu/). sgRNAs (gBruce #1 TGCATGCGCTGCGACGCCGA, gBruce #2 GCATGCGCTGCGACGCCGAC, gBRUCE#1 GCATGCACTGCGACGCCGAC, gBRUCE#2 TGCATGCACTGCGACGCCGA) were cloned into pSpCas9(BB)-2A-GFP (PX458) (a gift from Feng Zhang (Addgene plasmid # 48138)[55]. MEFs or HAP1 cells were transfected using GeneJuice transfection reagent (VWR International) and FACS sorted for GFP+ cells 48 h post transfection. A pooled cell line was used for subsequent experiments and compared to a mock transfected (pSpCas9(BB)-2A-GFP empty vector) cell line.

**Retroviral infection**. A method for retrovirus production is described elsewhere[56]. Briefly, retroviral plasmid and helper plasmid were transfected in packaging cell lines, Plat-E (Ecotropic) or Plat-A (Amphotropic), using a standard calcium-phosphate transfection protocol. 48 h post transfection, filtered condition media

**Fig. 6** BRUCE interacts with an autophagosome-lysosome fusion regulator Syntaxin 17. **a** Co-immunoprecipitation of GFP-Syntaxin 17 (STX17) with Myc-BRUCE. GFP-STX17 and Myc-BRUCE WT were transfected in HEK293T cells. Myc-BRUCE was immunoprecipitated using anti-Myc antibody from total cell lysates. Myc-BRUCE and GFP-STX17 were detected by western blotting, using the indicated antibodies. *Nonspecific band. **b** Confocal microscopy images of Bruce[+/+] and Bruce[−/−] MEFs, stably expressing exogenous GFP-STX17. Cells were grown in fully supplemented medium (FM) or starved for 2 h (Starv) with or without Bafilomycin (Baf; 100 nM). Scale bars, 20 μm. **c** Colocalization analysis of endogenous LC3B and exogenously expressed GFP-STX17. Bruce[+/+] and Bruce[−/−] MEFs stably expressing GFP-STX17 were grown in FM or starved for 2 h with or without Bafilomycin (100 nM), fixed, and stained for endogenous LC3B. Line plots across 8 μm distance (marked with arrows) exemplify degree of colocalization of GFP-STX17 and LC3B signals. Scale bars, 20 μm

containing retroviral particles supplemented with 4 μg ml$^{-1}$ polybrene (Sigma, H9268) were used to infect MEFs. To obtain stable cell lines, cells were selected using 1.5 mg ml$^{-1}$ G418 (Gibco, 108321-42-2), or 4 μg ml$^{-1}$ puromycin (Lactan GmbH, 240.3).

**Flow cytometry-based screening using an shRNA library.** A method to generate shRNA library is described elsewhere[45,52,53]. Briefly, oligomers were amplified by PCR attaching miR-E compatible overhangs, and cloned into pRSF91-SFFV-TagBFP-mirE-PGK-Neo-WPRE using EcoRI/XhoI restriction sites. Subsequently,

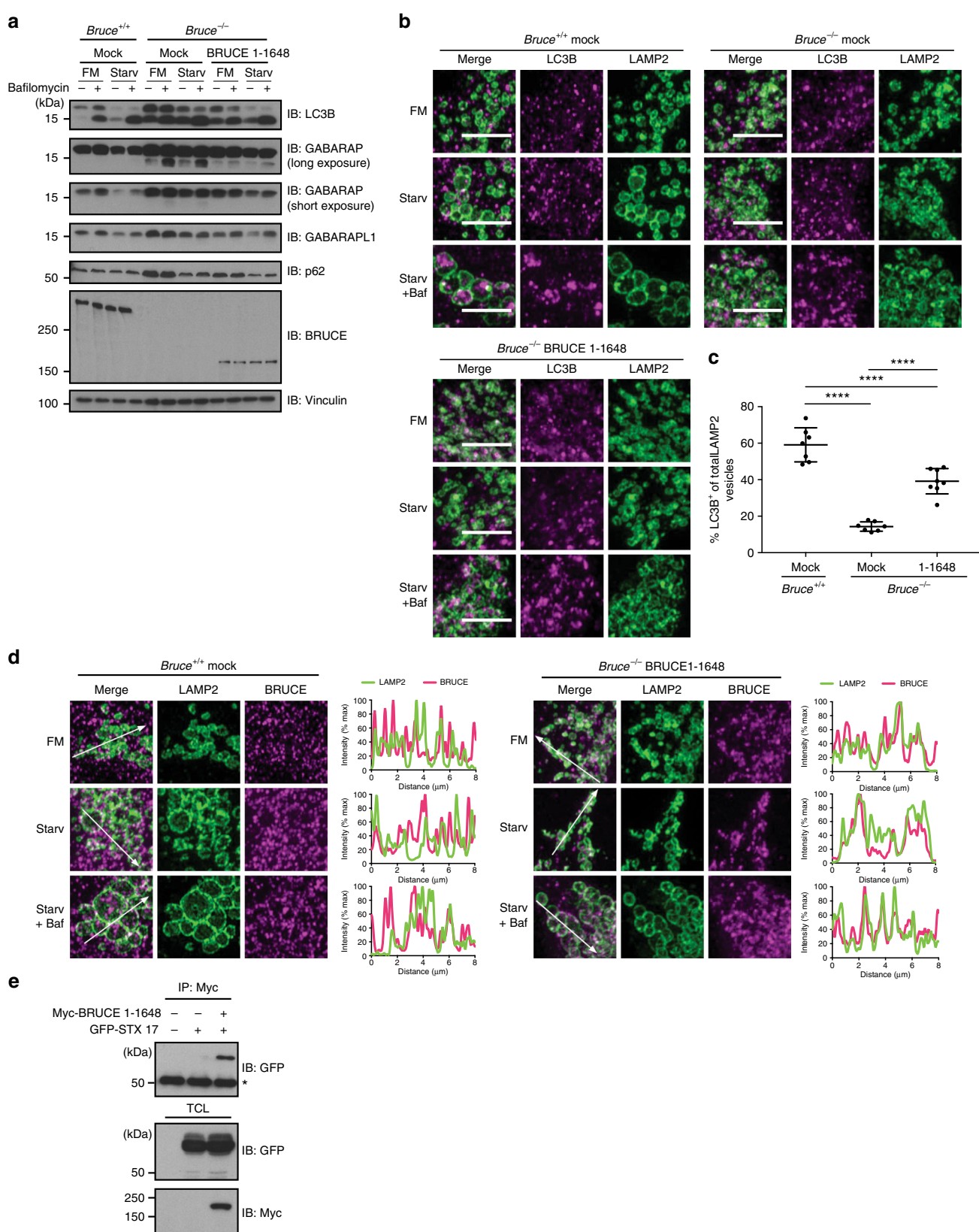

they were transformed into electrocompetent MegaX DH10B T1 (Invitrogen, C6400-03) in a scale to reach 10,000× representation of each shRNA. mCherry-EGFP-LC3B MEF line (Clone 1) was test infected with different retrovirus dilutions to determine transduction efficiency for single cell infection. To obtain 4.184 million transduced cells (1,000 cells per shRNA), at a determined transduction efficiency of 8.5%, 50 million MEFs were seeded in duplicates. The following day, the cells were infected by the calculated amount of virus and selected by G418 (1.5 mg ml$^{-1}$) after 2 days of infection. On day 8 after infection, cells were starved in Earle's balanced salt solution (EBSS, Life Technologies) for 6 h. Based on the GFP signal intensity, "GFP high" and "GFP low" population were sorted by FACS sorting. Genomic DNA was extracted using a standard phenol extraction protocol, followed by Proteinase K digestion. Subsequently, the guide strand of every shRNA was amplified by PCR while attaching sample-specific barcodes, keeping 1,000× representation of each shRNA (27 µg PCR template for 4,184 shRNAs). PCR samples were purified and sequenced using Illumina Solexa NGS (Next Generation Sequencing Facility, VBCF, Vienna, Austria). For the analysis of the NGS results, the normalized reads in the "GFP high" population were divided by normalized reads in the "GFP low" population, to obtain the fold change value for every shRNA. Subsequently, the geometric mean of both replicates was calculated, whereas scoring shRNAs are considered to have a geometric mean showing at least three-fold enrichment. For the hit list analysis only genes with at least three scoring shRNAs were considered. The average of the geometric mean of scoring shRNAs is displayed.

**Autophagic flux determination by FACS.** MEFs stably expressing mCherry-EGFP-LC3B were seeded in triplicates and cells were starved for 6 h in EBSS on the following day. After 6 h of starvation, cells were trypsinized, centrifuged (220 g, 5 min), and resuspended in 300 µl phosphate-buffered saline (PBS). Samples were analyzed for mCherry/GFP fluorescence signal using FACS Fortessa (BD). For mitophagy induction, cells were treated for 24 h with 1 µM Antimycin A (Sigma-Aldrich, A8674-25MG, 10 mM in dimethyl sulfoxide (DMSO)) and Oligomycin (Sigma-Aldrich, O4876-5MG; 10 mM in DMSO). Vehicle control was taken by using equal amounts of DMSO. For the analysis of basal fluorescence levels, the mean of the relative fluorescence units was displayed. To display autophagy induction, the fluorescence level in full medium or DMSO control condition was set to 100% for every cell line and the remaining fluorescence signal after autophagy induction was calculated accordingly.

**Fluorescence microscopy.** Cells were seeded on cover slips at low density. On the following day, cells were starved in EBSS, and if indicated treated with 100 nM Bafilomycin A1 (Enzo Life Sciences, BML-CM110-0100). Cells expressing fluorescently tagged proteins were fixed for 15 min at room temperature in 4% paraformaldehyde, washed, and subsequently transferred to mounting medium (VECTASHIELD, Szabo-Scandic, VECH-1000 (no DAPI) or VECH-1200 (containing DAPI)). For staining of endogenous proteins, cells were fixed in cold methanol for 10 min at −20 °C, re-hydrated in cold PBS on ice, followed by three washes. Cover slips were blocked in 5% bovine serum albumin (BSA) in PBS for 1 h at room temperature. Primary antibody was diluted in blocking solution according to the manufacturer's recommendations and incubated overnight on cover slips. Samples were washed in PBS and incubated in secondary antibody, Alexa Fluor 568 goat anti-mouse IgG (H+L) (Invitrogen, A11031), or Alexa Fluor 488 goat anti-rat IgG (H+L) (Invitrogen, A11006), diluted 1:1,000 in blocking solution for 1 h at room temperature. Samples were washed and cover slips were transferred to mounting medium. Samples were imaged by confocal microscopy or Airy scan on LSM780 or LSM880 Axio Observer (Zeiss). The analysis of mCherry-EGFP-LC3B-labeled vesicles was performed on Airyscan 3D image stacks using Definiens Developer Software. Cell borders were defined on a down-sampled image. A Bandpass filter was used on both channels for shaping out vesicles and a threshold was applied. The resulting objects were reshaped and tested for parameters like

mean intensity and coefficient of variation to filter out false positive ones. Spot volumes, intensities, and overlap of the two channels were measured. More than 40% overlap were regarded as double positive. For the analysis of LAMP2-positive vesicles, the diameter, number, and distance to the nucleus was manually quantified using Fiji software. For the quantification of LC3B containing LAMP2 vesicles, the number of strongly stained LC3B aggregates was counted manually using Fiji.

**Lysosomal pH-based assays.** To monitor changes in lysosomal pH, in the last 30 min of autophagy inducing treatment time, the growth medium was changed to 50 nM LysoTracker Red DND-99 (Invitrogen, L7528) containing medium. Cells were washed, trypsinized, and analyzed by flow cytometry. For microscopic analysis of LysoTracker Red stained cells, the growth medium was changed to 500 nM dye containing medium 30 min before the end of treatment time. Cells on cover slips were then washed for 10 min in chilled dye-free medium and fixed in 4% paraformaldehyde PBS, for 20 min on ice. Cover slips were washed once with chilled PBS and transferred to mounting medium and analyzed using LSM780 Axio Observer (Zeiss).

**Lysosomal enzymatic activity assay.** To monitor changes in lysosomal enzymatic activity, cells were starved for 2 h in EBSS and subsequently incubated in EBSS containing 10 µg ml$^{-1}$ DQ-Ovalbumin (Thermo Fisher Scientific, D12053) for 0, 15, 30, and 60 min. Cells on cover slips were washed, fixed for 15 min in 4% paraformaldehyde-PBS, and transferred to mounting medium containing DAPI nuclear staining. Samples were analyzed using LSM780 Axio Observer (Zeiss).

**Electron microscopic analysis.** MEFs were grown on 12 mm Aclar plastic discs and fixed for 1 h in 2.5% glutaraldehyde in 0.1 M sodium phosphate buffer, pH 7.4. Samples were then rinsed with the same buffer, subsequently fixed in 1% osmium tetroxide in ddH$_2$O, dehydrated in a graded series of ethanol and embedded in Agar 100 resin. Seventy-nanometer sections were cut parallel to the substrate and post-stained with 2% uranyl acetate and Reynolds lead citrate. Sections were examined with an FEI Morgagni 268D (FEI, Eindhoven, The Netherlands) operated at 80 kV. Images were acquired using an 11-megapixel Morada CCD camera (Olympus-SIS). The number of autophagic bodies per autolysosome was manually quantified.

**Immunoblotting.** The method is described elsewhere[46]. Briefly, cells were lysed with chilled lysis buffer (50 mM HEPES, 150 mM NaCl, 1 mM EDTA, 1 mM EGTA, 25 mM NaF, 10 mM ZnCl$_2$, 10% glycerol, 1% Triton X-100, 20 mM NEM, and Complete protease inhibitors) and total cell lysates were resolved by SDS-PAGE, and transferred to nitrocellulose (GE Healthcare, Little Chalfont, UK) or to polyvinylidene difluoride membrane (Millipore, ISEQ00010). Membranes were blocked in 5 % BSA-TBS and blotted with indicated antibodies in blocking solution at 4 °C overnight. The following secondary antibodies were used according to the manufacturer's recommendations: goat anti-mouse HRP (BioRad, 170–6516; 1:7,000), goat anti-rabbit HRP (Dako, P0448; 1:2,000), and goat anti-rat IgG HRP (Southern Biotech, 3050-05; 1:10,000). Western blotting Luminol Reagent (Santa Cruz) and high-performance chemiluminescence films (GE Healthcare) were used. Where appropriate, Ponceau S staining was used to visualize transferred proteins on the membranes.

For Odyssey-based immunoblotting quantification, membranes were blocked in 5% milk PBS and washed using PBS 0.2% Tween. Primary and secondary antibodies were diluted according to the manufacturer's recommendations in 5% milk PBS 0.2% Tween. IRDye 800CW (925–32210; 1:10,000) and IRDye 680RD (925–68071; 1:10,000) secondary antibodies were purchased from LICOR. Membranes were imaged and quantified using LICOR imaging system. LC3B levels were subsequently normalized to Vinculin loading control.

**Fig. 7** BRUCE (aa 1–1648) partially rescues the autophagy defect in *Bruce*$^{-/-}$ MEFs. **a** Protein levels of LC3B, GABARAP, GABARAPL1, and p62 in *Bruce*$^{-/-}$ MEFs stably expressing BRUCE 1–1648 compared with *Bruce*$^{+/+}$ and *Bruce*$^{-/-}$ empty vector (mock) expressing MEFs. Total cell lysate of MEFs in fully supplemented medium (FM), starved for 2 h (Starv) with or without Bafilomycin A1 (Baf; 100 nM) was analyzed by immunoblotting using antibodies as indicated. Vinculin was monitored as a loading control. **b** Confocal microscopy images of LC3B and LAMP2 in *Bruce*$^{+/+}$ (mock) and *Bruce*$^{-/-}$ (mock and BRUCE 1–1648) MEFs. MEFs in basal or 4 h starved condition with or without Bafilomycin A1 (100 nM) were fixed and stained for endogenous LC3B and LAMP2 using antibodies as indicated. Scale bars, 5 µm. **c** Quantification of LAMP2-positive vesicles containing LC3B-positive aggregates based on microscopy images from (**b**) in 4 h starved condition treated with Bafilomycin A1. Data are presented as dot plots with mean±SD (****$p < 0.0001$). $n = 8$ cells. **d** Colocalization analysis of LAMP2 and BRUCE in *Bruce*$^{+/+}$ and *Bruce*$^{-/-}$ mock compared to BRUCE 1–1648 expressing MEFs. MEFs in basal, 4 h-starved condition with or without Bafilomycin A1 (100 nM) were fixed and stained for endogenous BRUCE and LAMP2 using antibodies as indicated. Fluorescent intensity of LAMP2 and BRUCE signals across 8 µm regions marked with arrows is shown in line plots. **e** Interaction of GFP-STX17 and Myc-BRUCE (aa 1–1648) determined by co-immunoprecipitation. GFP-STX17 and Myc-BRUCE (aa 1–1648) were transfected in HEK293T cells. Myc-BRUCE (aa 1–1648) was immunoprecipitated using anti-Myc antibody from total cell lysates (TCL). Immunoprecipitated samples were examined by immunoblotting using the indicated antibodies. *Nonspecific band

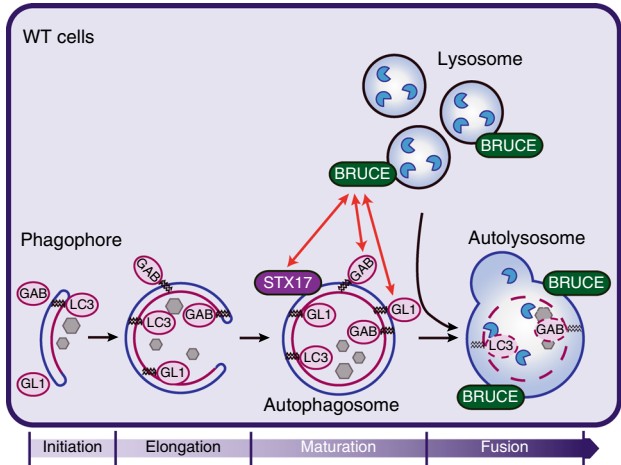
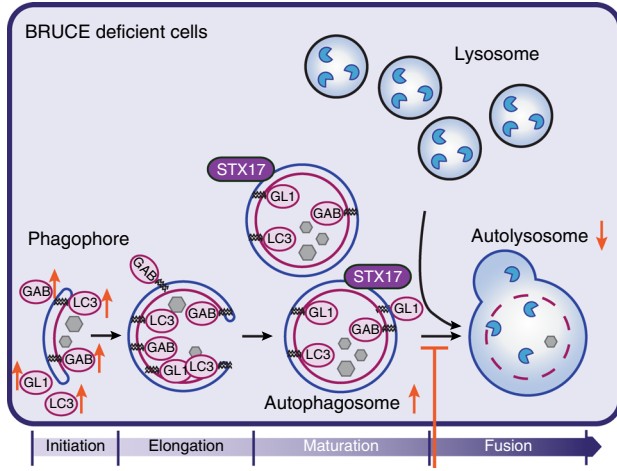

**Fig. 8** BRUCE regulates autophagy flux by promoting autophagosome-lysosome fusion. Schematic illustrating the functional role of BRUCE during autophagy induction in wild type cells (left panel) and in BRUCE deficient cells (right panel). BRUCE interacts with GABARAP, GABARAPL1, and STX17, and localizes at the lysosomal as well as autolysosomal membranes (left panel). BRUCE-deficient cells have reduced numbers of autolysosomes and display an accumulation of LC3B, GABARAP, and GABARAPL1 due to inefficient autophagosome-lysosome fusion (right panel). LC3 (LC3B), GAB (GABARAP), GL1 (GABARAPL1), STX17 (Syntaxin 17)

Uncropped blots from main figures can be found in supplementary figures 8 – 12.

**EGF receptor degradation assay**. Cells were stimulated with 100 ng ml$^{-1}$ EGF (Thermo Scientific, E13345) in starvation medium after washing with PBS twice. At the indicated time points, cells were washed with chilled PBS and lysed in lysis buffer. Lysate was centrifuged for 15 min, 13,000 $g$ at 4 °C and supernatant was used for immunoblotting.

**GST-protein purification and pulldown assay**. A method for GST-protein purification and pulldown assay are described elsewhere[46]. Briefly, proteins were expressed in *Escherichia coli* BL21 (DE3) overnight at 18 °C and purified by Glutathione Sepharose 4B agarose beads (GE Healthcare). Total cell lysates were subjected to GST pulldown at 4 °C overnight using agarose-beads immobilized GST-control or GST-tagged proteins as indicated. After washing with chilled lysis buffer for 3 times, the pulldown samples were subjected to SDS-PAGE followed by immunoblotting. Input of the GST-tagged proteins or GST-control was analyzed by Ponceau S staining.

**Immunoprecipitation**. Myc-BRUCE was immunoprecipitated from HEK293T cells, 48 h after transfection with the indicated plasmids using GeneJuice transfection reagent. Total cell lysates from 10 cm dishes were prepared as mentioned above, and incubated with 4 µg anti-Myc (9E10, Covance, MMS-150P) for 7 h at 4 °C, followed by incubation with 40 µl Protein G agarose beads (Sigma, 11243233001) for 1 h. Beads were washed four times, resuspended in loading buffer and boiled at 96 °C for 5 min. Samples were analyzed by immunoblotting.

**RNA sequencing**. To prepare RNA-Seq samples, total RNA was isolated from MEFs expressing *shRenilla* and *shBruce*#1, starved for 2 h or grown in regular media using TRIzol (Life Technologies, 15596026). Contaminating DNA was digested by TURBO DNA-free Kit (Ambion, AM1907) and Bioanalyzer 2100 (Agilent Technologies) was used to determine the quality and quantity of RNA according to the manufacturers' instructions. The library was prepared from these samples by poly(A) enrichment (New England Biolabs, Ipswich, MA). The resulting fragmented samples were sequenced on a HiSeqV4 SR50 with a read length of 50 (by VBCF-NGS). The reads were mapped to the *Mus musculus* mm10 reference genome with STAR (version 2.4.0d)[57]. Reads aligning to rRNA sequences were filtered out prior to mapping. The read counts for each gene were detected using HTSeq (version 0.5.4p3)[58]. The counts were normalized using the TMM normalization from edgeR package in R. Before statistical testing, the data was voom transformed and then the differential expression between the sample groups was calculated with limma package in R. The functional analyses were done using the topGO and gage packages in R. For visualization, heat maps were created using R and fragment alignments were processed using the Integrative Genomics Viewer (IGV_2.3.40 software)[59,60].

**Statistical analysis**. All graphs were created using GraphPad Prism 7 software (GraphPad Software, Inc). The ANOVA test was used for all data sets, excluding indicated data sets in which two groups were compared by $t$-test. Significance and confidence level was set at 0.05.

**Data presentation**. Representative western blots are shown from three independent experiments in Figs. 1d, e, 2d, 3a, b, 6a and 7a, e, S3b, S3d-f, S4a-h, S4o, S6c, S7a-f and S7i; from two independent experiments in Figs. 2f and 3c, S3g-h, S4i-l, S4n, S6d and S7g–h. Representative confocal microscopy images are displayed from three independent experiments in Figs. 2b, 5a, 6b, 7b and d, S1e, S5a and S6a; from two independent experiments in Figs. 4a and 5d and S6b; from one experiment in Fig. 6c. Representative electron microscopy images in Fig. 5c are taken from two independent experiments. Representative replicate#1 from the shRNA screen is shown in Fig. 1b and Fig. S2b. Corresponding correlation plots for both replicates are displayed in Fig. S2c. Representative FACS plots from two independent experiments from Fig. S1b–c are shown in Fig. S1d. Odyssey-based quantification of Fig. S3c was taken from one experiment using identical samples as in Fig. 7a. Specific data sets from RNA-Seq experiment are displayed in Fig. S3i and S5d.

All assays used in this study were pre-established, and planned accordingly or variance was determined from pilot studies. Statistical method was not used to determine the power analysis.

No samples were excluded from the analysis, and no specific statistical method was used for randomization. Mean-variance relationship is estimated empirically.

For quantification of microscopy data, the investigator was not blinded.

**Data availability**. The RNAseq data that support the findings of this study have been deposited to GEO Submission with the GSE102808 accession code in the NCBI tracking system #18600616. Rest of the data that support the findings of this study are available from the corresponding author upon reasonable request.

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

## Acknowledgements

We acknowledge Barbara Hopfgartner and Thomas Gstrein (IMP, Vienna, Austria) for technical advice and Tim Clausen (IMP) for his advice on the protein folding prediction. We also thank Haplobank (IMBA, Vienna, Austria) for haploid cells, Molecular Biology Service and BioOptics, especially Thomas Lendl for developing an automated program to analyze autophagic vesicles, Alex Schleiffer for bioinformatics analysis (IMP-IMBA core facilities), Sini Junttila for RNA sequencing data analysis (BioComp, VBCF, Vienna, Austria), Nicole Fellner and Thomas Heuser for EM analysis (Electron Microscopy Facility, VBCF), and Next Generation Sequencing Core Facility (VBCF) for the sample processing. Projects in Ikeda Lab are supported by the ERC consolidator grant (LUbi, 614711), the FWF stand alone grant (P 25508), OeNB, COST (European Cooperation in Science and Technology, PROTEOSTASIS BM1307), and Austrian Academy of Sciences. Research in Zuber lab is supported by the ERC starting grant (ChromatinTargets). We thank Yasin F Dagdas (GMI, Vienna, Austria), Claudin Kraft (MFPL, Vienna, Austria) and all the members in Ikeda Lab for scientific discussions on the project, Sascha Martens (MFPL) for the critical reading and comments on the manuscript, and Life Science Editors for editorial assistance.

## Author contributions

P.E., I.P., L.D. and T.H. carried out the experiments and analyzed the data. P.E., J.Z. and F.I. analyzed and interpreted the data. J.Z. and F.I. designed the experiments. F.I. wrote the paper. All the authors contributed to finalize the manuscript.

## Additional information

**Competing interests:** The authors declare no competing financial interests.

