## [Peer Review File · Nature Communications]

Editorial Note: this manuscript has been previously reviewed at another journal that is not operating a transparent peer review scheme. This document only contains reviewer comments and rebuttal letters for versions considered at *Nature Communications*.

REVIEWERS' COMMENTS:

Reviewer #1 (Remarks to the Author):

I have reviewed the revised manuscript and response to my previous critique by Ebner et al. The authors have adequately addressed my concerns, have improved this manuscript and I feel it is appropriate to publish this in Nature Communications.

Reviewer #2 (Remarks to the Author):

Although the authors tried to resolve them, most of previous concerns have not been addressed satisfactorily. However, in the light of interest of their finding to the field, I feel that the revised paper would be of value to the general readership of Nature communications.

We thank both reviewers for taking time to comment on our revised manuscript.

REVIEWERS' COMMENTS:

Reviewer #1 (Remarks to the Author):

I have reviewed the revised manuscript and response to my previous critique by Ebner et al. The authors have adequately addressed my concerns, have improved this manuscript and I feel it is appropriate to publish this in Nature Communications.

We are very happy to hear about the positive comments from the reviewer about our revised manuscript to be appropriate for the publication in Nature Communications.

Reviewer #2 (Remarks to the Author):

Although the authors tried to resolve them, most of previous concerns have not been addressed satisfactorily. However, in the light of interest of their finding to the field, I feel that the revised paper would be of value to the general readership of Nature communications.

Indeed, there remain open questions, which we were not able to address entirely. However, we are thankful for the reviewer's comment on our revised manuscript to be of value to the general readership of Nature Communications.